# Explore Long-Range Context Features for Speaker Verification

Zhuo Li [1,2], Zhenduo Zhao [1,2], Wenchao Wang [1], Pengyuan Zhang [1,2] and Qingwei Zhao [1,2,*]

1 Key Laboratory of Speech Acoustics and Content Understanding, Institute of Acoustics,
  Chinese Academy of Sciences, Beijing 100190, China
2 University of Chinese Academy of Sciences, Beijing 100049, China
* Correspondence: qzhao@hccl.ioa.ac.cn

**Abstract:** Multi-scale context information, especially long-range dependency, has shown to be beneficial for speaker verification (SV) tasks. In this paper, we propose three methods to systematically explore long-range context SV feature extraction based on ResNet and analyze their complementarity. Firstly, the Hierarchical-split block (HS-block) is introduced to enlarge the receptive fields (RFs) and extract long-range context information over the feature maps of a single layer, where the multi-channel feature maps are split into multiple groups and then stacked together. Then, by analyzing the contribution of each location of the convolution kernel to SV, we find the traditional convolution with a square kernel is not effective for long-range feature extraction. Therefore, we propose cross convolution kernel (cross-conv), which replaces the original $3 \times 3$ convolution kernel with a $1 \times 5$ and $5 \times 1$ convolution kernel. Cross-conv further enlarges the RFs with the same FLOPs and parameters. Finally, the Depthwise Separable Self-Attention (DSSA) module uses an explicit sparse attention strategy to capture effective long-range dependencies globally in each channel. Experiments are conducted on the VoxCeleb and CnCeleb to verify the effectiveness and robustness of the proposed system. Experimental results show that the combination of HS-block, cross-conv, and DSSA module achieves better performance than any single method, which demonstrates the complementarity of these three methods.

**Keywords:** speaker verification; long-range dependencies; hierarchical-split block; cross convolution; depthwise separable self-attention module





## 1. Introduction

Speaker verification (SV) aims to identify or verify an individual's identity from samples of his/her voice, which has developed rapidly over the years. SV systems based on neural networks have emerged in recent years and achieved state-of-the-art performance. SV systems consist of two parts, (i) a front-end, aka feature extractor, that extracts speaker embedding and (ii) a back-end that produces a likelihood ratio score using the enrollment and test embeddings. Front-ends generally consist of three parts, one frame-level speaker feature extractor (hereinafter referred to as feature extractors), one pooling layer for statistics extraction and one embedding extractor for segment-level speaker embedding extraction.

Various neural network feature extractors are key ingredients to achieving great improvements in SV tasks. D-vector [1] firstly proposes a neural network feature extractor. X-vector [2] and ResNet [3–6] are two of the most popular feature extractors in the past years. X-vector uses the Time Delay Neural Network (TDNN) as a feature extractor to consider the dependencies of contiguous frames, and ResNet utilizes the convolutional neural network as frame-level feature extractors to model speaker information. While, because the TDNN or CNN units can not model dependencies outside the receptive field, it is challenging to model long-range dependencies due to the limited receptive fields (RFs). In addition, speech has a rich temporal structure over multiple timescales, and identity-relevant acoustic information is carried on more than one timescale, such as 10–30 ms for fundamental frequency, 100–300 ms for rhythm, and more than 1000 ms for

intonation information. Meanwhile, numerous biological researches [7,8] show that the brain displays a complex multiscale temporal organization to track acoustic dynamics of different timescales. Thus, enlarging the model receptive field is crucial for SV task. Here, TDNN can be regarded as a one-dimensional convolution; thus, CNN is used in the following paper for convenience.

The most straightforward method is to deepen the network by stacking neural network layers, e.g., DTDNN [9], ResNet101. However, there are two important problems. First, The features extracted from the serial network generated by stacking layers contain single-scale speaker information. Second, study [10] shows that the empirical RFs gained by a chain of convolutions are much smaller than the theoretical RFs by experiments, especially in deeper layers. Another work [11] has proven that the distribution of impact within an effective receptive field is limited to a local region and converged to the gaussian. Therefore, speaker features extracted by the serial network are mostly localized, i.e., short, due to the limited receptive field and model parameters.

Recently, Res2Net, which was proposed in [12], enlarges the model receptive field and extracts multi-scale features by splitting channels into multiple groups and stacking them in a single layer. Res2Net module is quickly introduced in the SV field and achieves great improvements. Quickly, this module is integrated into many SV backbone [13]; ECAPA is the most famous backbone. In addition, some modified structures are also proposed, such as SI-Net [14], HS-ResNet [15].

In addition to these methods, enlarging the receptive field of a single layer is another straightforward method, and we explore it in our prior work [14]. We utilize one large receptive field branch to extract large-scale features and explore the complementarity between large-scale features and normal-scale features. Similarly, studies [16–19] shows that adopting larger kernel-sized convolutions achieves significant performance improvements in the computer vision field. However, the inference time is intolerable, and the model is prone to overfitting during training.

The above methods all extract long-range information from a local perspective; other methods extract long-range information by modeling the pairwise relations globally based on LSTM and the attention mechanism [20]. In the early years, some studies [21–24] insert the LSTM on the top of or inside the backbone network for SV tasks to model long-range information because the LSTM is good at modeling the long-term information. The authors in [25] combine the BLSTM and ResNet into one unified architecture to model long-range context information. Recently, the Long-Short Range Attention module [26], a multi-branch feature extractor, is proposed and utilizes attention to concentrate on global context. Conformer [17] redesigns the combination of convolution and attention to take full advantage of the global information extracted from speech. Moreover, numerous global-based approaches [27–29] based on attention mechanisms have been explored to promote effective long-range interactions in computer vision (CV) and the speech field.

Overall, these studies indicate that there are three main effective approaches to extracting long-range context features, stacking layers, enlarging the receptive field of a single layer, and using the attention mechanism. Thus, this paper has four key aims. First, we introduce a Hierarchical-Split [15] block to enlarge the RFs in a single block for SV tasks by splitting multi-channel feature maps into multiple groups and stacking them, named HS-ResNet. Second, to alleviate the problem of a sharp increase in inference time and overfitting caused by large convolution kernels, we analyze the contribution of each location of the convolution kernel to speaker verification by removing some weights at different spatial locations and observing performance degradation using ResNet34 on VoxCeleb1. After that, we find the traditional convolution with a square kernel is ineffective for long-range feature extraction and propose cross-convolution kernels (cross conv) for speaker verification. Third, we propose the DSSA module, a plug-and-play module based on the attention mechanism that can be easily plugged into various existing architectures to improve performance. Fourth, we explore the complementarity of these three approaches.

We combine these three methods on a single model for comparative experiments to explore complementarity.

The major contributions are summarized as follows.

- we propose cross-conv to enlarge RFs locally by using the pruning method and the idea of removing local parameters to evaluate the importance of the location.
- we propose an innovative plug-and-play module based on the attention mechanism, DSSA module. DSSA module is flexible and extendable, and it can easily be plugged into multiple mature architectures to improve performance.
- We explore the long-range context information extracted by these three methods by combining them into one model.

The organization of this paper is as follows. The proposed HS-block is described detailedly in Section 2. Section 3 demonstrates the DSSA module, which can be taken as a plug-and-play module based on the attention mechanism. Cross-conv is explored in Section 4. The experiment settings, results, and analysis are given in Section 5. Section 6 concludes the paper.

## 2. Related Work

### 2.1. Speaker Verification System Overview

As introduced in Section 1, two-stage speaker verification systems based on neural networks, which consist of a front-end and a back-end, are the predominant approach. As shown in Figure 1, the front-end, that is, the feature extractor, that extracts speaker embeddings from utterances, comprises three key components, a frame-level feature extractor, a pooling layer, and a segment-level feature extractor. The frame-level feature extractor uses CNN and TDNN to extract frame-level speaker information, the pooling layer converts frame-level features to segment-level features, and the utterance-level network further maps the utterance-level features to obtain speaker embeddings. In the training stage, the parameters of the feature extractor are optimized by using the softmax loss function. While in the test stage, speaker embeddings generated by the feature extractor are used to calculate the similarity between the enrollment utterance and the test utterance.

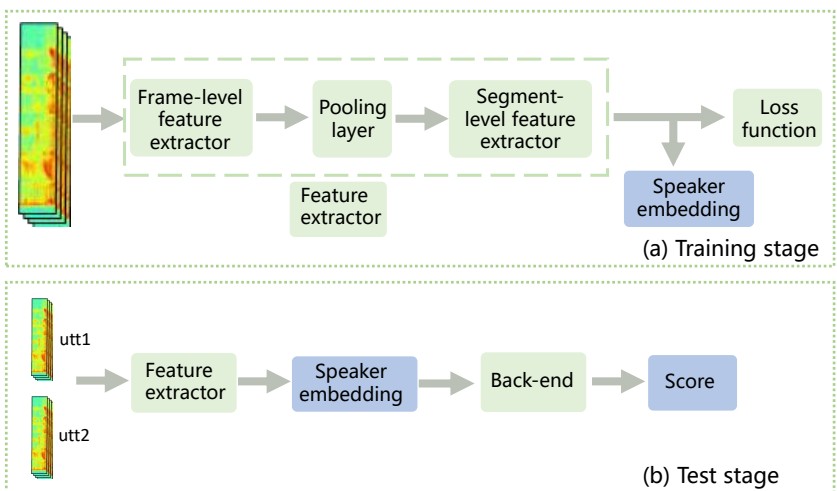

**Figure 1.** Speaker verification system diagram.

X-vector [2] and ResNet [3] are two of the most representative speaker feature extractors. X-vector employs a multi-layer TDNN structure, which models temporal context information through the time-delay units. In contrast, ResNet uses stacked convolutional layers and incorporates a residual connection to mitigate the vanishing gradient problem caused by deep neural network architectures. Based on the x-vector, various TDNN architectures have been proposed, such as ETDNN [30], FTDNN [31] and D-TDNN [9], which improve performance by deepening the network and enhancing the temporal con-

text modeling capability. The FTDNN architecture, in particular, employs singular value decomposition to reduce the number of parameters in the model. Furthermore, the ECAPA architecture utilizes inter-layer channel splitting and stacking to enhance its modeling capabilities. One of the most well-known variations of ResNet is Res2Net [12], which splits the multi-group feature maps into multiple groups and stacks them together, resulting in substantial improvements in speaker verification tasks.

### 2.2. ResNet

ResNet, proposed by [3], is one of the most popular architectures for speaker verification tasks due to its impressive performance. It consists of stacked convolutional layers divided into four stages (marked with stages 1/2/3/4). The ResNet34 and ResNet50 structures are among the most widely used, and the detailed configures are shown in Figure 2. Each stage contains multiple ResNet blocks, each of which consists of several convolutional layers and a shortcut connection from the input to the output. In addition, $s = 2$ means that this stage includes a down-sampling operation through the use of a stride of 2 in the first convolutional layer. As shown in Figure 2, the output of this stacked structure is fed into a pooling layer and a fully connected layer to obtain segment-level embedding. Finally, the output of the fully-connected layer, known as the embedding, is used to calculate loss and optimize classification during training and to calculate the score during testing.

| Layer name | ResNet34 | ResNet50 | Output size |
|---|---|---|---|
| Input | — | — | $B \times 1 \times T \times F$ |
| Conv | $3 \times 3, C$ | $3 \times 3, C$ | $B \times C \times T \times F$ |
| Stage1 | $\begin{bmatrix} 3 \times 3, C \\ 3 \times 3, C \end{bmatrix} \times 3$ <br> $s = 1$ | $\begin{bmatrix} 1 \times 1, C \\ 3 \times 3, C \\ 1 \times 1, C \end{bmatrix} \times 3, s = 1$ | $B \times C \times T \times F$ |
| Stage2 | $\begin{bmatrix} 3 \times 3, 2C \\ 3 \times 3, 2C \end{bmatrix} \times 3$ <br> $s = 2$ | $\begin{bmatrix} 1 \times 1, 2C \\ 3 \times 3, 2C \\ 1 \times 1, 2C \end{bmatrix} \times 3, s = 2$ | $B \times 2C \times \frac{T}{2} \times \frac{F}{2}$ |
| Stage3 | $\begin{bmatrix} 3 \times 3, 4C \\ 3 \times 3, 4C \end{bmatrix} \times 3$ <br> $s = 2$ | $\begin{bmatrix} 1 \times 1, 4C \\ 3 \times 3, 4C \\ 1 \times 1, 4C \end{bmatrix} \times 3, s = 2$ | $B \times 4C \times \frac{T}{4} \times \frac{F}{4}$ |
| Stage4 | $\begin{bmatrix} 3 \times 3, 8C \\ 3 \times 3, 8C \end{bmatrix} \times 3$ <br> $s = 2$ | $\begin{bmatrix} 1 \times 1, 8C \\ 3 \times 3, 8C \\ 1 \times 1, 8C \end{bmatrix} \times 3, s = 2$ | $B \times 8C \times \frac{T}{8} \times \frac{F}{8}$ |
| Pooling & Flatten | Average pooling | | $B \times 8C \times 1 \times \frac{F}{8}$ <br> ➜ $B \times (F \cdot C)$ |
| Fc layer | Fully connected layer | | $B \times 512$ |
| Final | Loss function / Scoring | | —— |

**Figure 2.** ResNet34 and ResNet50 architecture.

## 3. Hierarchical-Split Block

Local-based approaches enlarge the local receptive field through pooling, dilated, split-stack and other operations usually. A hierarchical-Split block is one of the approaches by split-stack operations.

The structure of the HS-block is depicted in Figure 3b. The $3 \times 3$ block in the typical ResNet is modified. After the $1 \times 1$ convolution, the feature maps are split equally into $s$ groups, denoted by $x_i$, and the $3 \times 3$ convolution filters are also replaced by several groups, denoted as $\mathcal{F}_i()$. Each $x_i$ will be fed into $\mathcal{F}_i()$, and the output feature maps are denoted by $y_i$. Here, each $y_i$ is equally split into two sub-groups, denoted $y_{i,1}$ and $y_{i,2}$. Then, $y_{i,2}$ is concatenated with the following group $x_{i+1}$, and then sent into $\mathcal{F}_{i+1}()$. All $y_{i,1}$ are concatenated in the channel dimension as the output of the $3 \times 3$ convolution filters, denoted as $y_i$. Especially, each $y_{i,1}$ has different channels and RFs, and the more channels $y_{i,1}$ contains, the larger RFs are gained. In this manner, the feature maps could

contain detailed information and larger-range dependencies. $\oplus$ means two feature maps are concatenated in the channel dimension.

$$y_i = \begin{cases} x_i, & i = 1 \\ \mathcal{F}_i(x_i \oplus y_{i-1,2}), & 1 < i <= s \end{cases} \tag{1}$$

Moreover, the hyperparameters $s$ and $t$ are used to control the HS-ResNet's parameters and complexity. $s$ is the group the feature maps are divided into, and $t$ means how many times the number of channels will be expanded. $k$ means the size of convolution kernel, $w$ means the number of channels. The computational complexity of the HS-block can be calculated as follows.

$$c_0 = t \times w = t \times \frac{C}{s} \quad c_i = c_0 + \frac{c_{i-1}}{2} \tag{2}$$

$$\begin{aligned} PARAM &= k^2 \times (\sum_{i=0}^{s-2} c_i^2 + c_0^2) \\ &= c_0^2 \times (4s - \frac{29}{3} + 16 \times 2^{-s} - \frac{16}{3} \times 2^{-2s}) \\ &= C^2 \times (\frac{t}{s})^2 \times (4s - \frac{29}{3} + 16 \times 2^{-s} - \frac{16}{3} \times 2^{-2s}) \end{aligned} \tag{3}$$

Here, we denoted:

$$\beta = (\frac{t}{s})^2 \times (4s - \frac{29}{3} + 16 \times 2^{-s} - \frac{16}{3} \times 2^{-2s}) \tag{4}$$

The parameters of the traditional convolutional layer are $C^2$, and the parameters of the HS-block are $\beta \times C^2$. By controlling the parameters $t$ and $s$, $\beta$, that is, the parameters of the HS-block can be precisely designed.

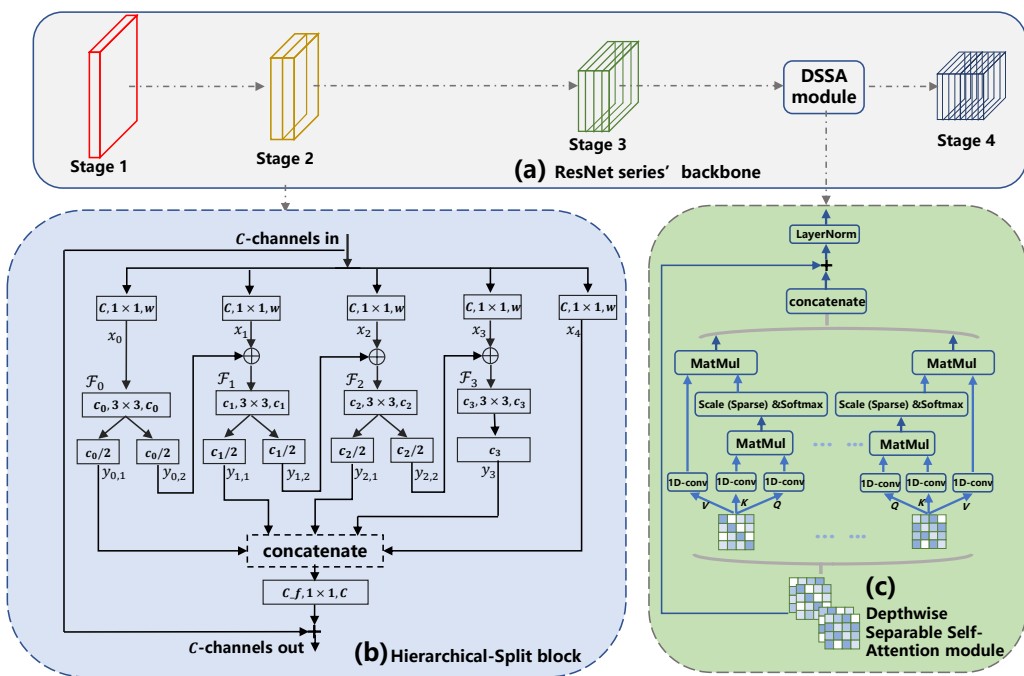

**Figure 3.** Detail structure of HS-ResNet50 and Depthwise Separable Self-Attention module for the combined system.

## 4. Depthwise Separable Self-Attention Module

The attention mechanism is generally used to model the pairwise relations from the global perspective. In this section, we introduce our proposed method, the DSSA module. Scaled Dot-Product Attention, a core module of the famous Transformer architecture, is

also the core of our module. Scaled Dot-Product Attention consists of three inputs: queries, keys, and values. The output of each query will be calculated by the dot products with outputs of all keys; the results are divided by $\sqrt{d_k}$ and applied a softmax function to obtain the weights of the values' output. Here, $d_k$ means the dimension of keys, and $d_q$ and $d_v$ are similar. In practice, the above calculations are implemented in the form of matrices. The outputs are computed as

$$Attention(Q, K, V) = softmax(QK^T / \sqrt{d_k})V \tag{5}$$

In the equation, $d_q = d_k = d_v = d_{in} = d$, the projection matrices $W^Q, W^K, W^V \in \mathbb{R}^{d \times d}$. $d_{in}$ means the dimension of the frequency axes of the input feature map.

Here are a few challenges when using Multi-Head Scaled Dot-Product Attention in CNN structures, especially the parameters. The depthwise separable convolution strategy, which decomposes ordinary convolutions into depthwise convolution and pointwise convolution, is introduced in the module to avoid the massive parameters.

In the DSSA module, the depthwise separable convolution applies a Scaled Dot-Product Attention to each input channel. As decipted in Figure 3c, the input feature maps $F$ are split into several groups, and the number of groups is equal to the number of channels $C$ in the feature maps, $F_i, i = 1, ..., C, F \in \mathbb{R}^{C \times T \times W}$. That is, each group only contains one channel of the feature maps. Then, each group is fed into three independent 1D-convolution to generate the queries, keys, and values, $Q_i, K_i, V_i$. The dimension of these is the same as the input. Specifically, the DSSA module inputs a $C \times T \times W$ feature map and produces three independent $C \times T \times W$ feature maps after the convolution layers.

$$Q_i, K_i, V_i = Conv1d_{Q,K,V}(F_i) \tag{6}$$

After that, the dot products between each query and each key are calculated and divided by $\sqrt{d_k}$. Before applying a softmax function to obtain the weights on the values', another square operation acts on the weights, which is to maintain weights in the normal range and avoid it being too big or too small.

$$\begin{aligned} D_i &= \sqrt{Q_i K_i^T / \sqrt{d_k}} \\ Y_i &= Softmax(D_i)V_i \end{aligned} \tag{7}$$

We perform the layer-norm operation on output of each group $Y_i$ in parallel. Finally, the outputs of groups are concatenated to get $y$.

$$y = LayerNorm(x + Y) \tag{8}$$

In addition, to facilitate the processing of speech with a long duration, the explicit sparse attention mechanism is designed in this module. The explicit sparse attention only pays attention to the nearest k frames. The attention weights degenerate to the sparse attention through k-nearest selection. The weights of k nearest frames of each row in the attention weight matrix are selected, and the others are replaced with $-\infty$.

$$\mathcal{M}(D_i, k)_{mn} = \begin{cases} D_{i,mn}, & if \ m \ge n - k/2 \ \& \ m \le n + k/2 \\ -\infty, & else \end{cases} \tag{9}$$

where $D_{i,mn}$ means the value of row $m$ and column $n$ in $i$ channel.

## 5. Cross-Convolution Kernel

Although enlarging the RFs of the convolution kernel is rather straightforward to extract long-range features, it causes problems of difficult training and slow inference. The main reason is parameters, and we propose a cross-convolution kernel to solve this problem.

Firstly, we analyze the contribution of each location of the traditional $3 \times 3$ convolution kernel to speaker verification by using the pruning method and the idea of removing local parameters to evaluate the importance. The training dataset and evaluation set are VoxCeleb1 and Vox1-O, and ResNet34 is used as a feature extractor. EER on Vox1-O and the accuracy of the training set are used to evaluate the performance degradation. The initial EER is 3.807, and the training accuracy is 99.99%.

In Figure 4a, we remove one weight of the convolution kernel. As we can see, the middle row is more important for speaker verification compared with other locations. Then, we remove two weights from the convolution kernel. As shown in Figure 4b, the white block represents the weight of this location is set to 0, and the weights of the orange block are reserved. Because removing the center of the convolution kernel causes significant performance degradation, we do not show these. Figure 4b shows that some patterns, especially (4) and (8), are not important for speaker verification, while patterns of (1), (3), (6) are important for speaker verification compared with others. These results show that the traditional convolution with a square kernel is redundant for speaker verification.

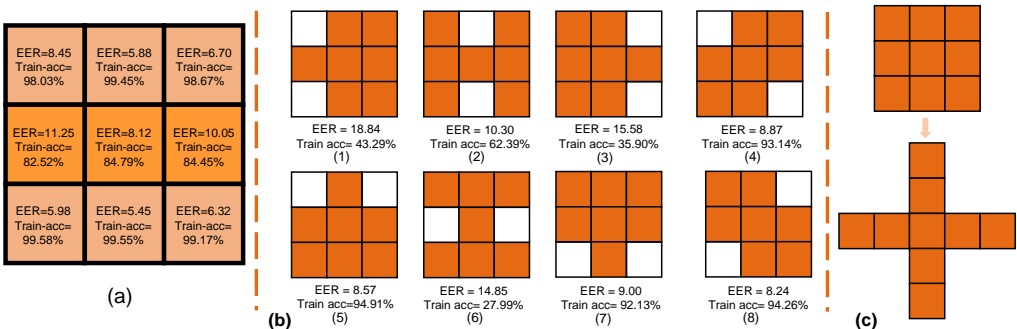

**Figure 4.** Performance degradation after removing local parameters.

Thus, considering the convenience and the importance of each location, we utilize $1 \times 5$ and $5 \times 1$ convolution kernels for replacing the original $3 \times 3$ convolution kernels and proposing cross-convolution kernels to enlarge RFs and enhance the ability of long-range feature extraction, as shown in Figure 4c. Because the output feature maps are calculated by sliding the convolution kernel over the input feature maps, the feature maps generated using cross-convolution contains all patterns except for (4) and (8).

## 6. Experiments

In this section, we first describe the datasets used for our experiments, VoxCeleb and CnCeleb. Then, the training details, including pre-processing, training strategies, and evaluation metrics, are presented.

### 6.1. Datasets and Augmentation

We conduct experiments on three datasets, VoxCeleb1&2 (Vox1&2) [32,33], and CnCeleb 1&2 [34,35].

The experiments are classified into two series according to the training set, the Vox series and the Cn series. The dev part of Vox1&2 are used as the training part of the Vox series, respectively, which contains 1211 speakers and 5994 speakers separately. Data augmentation is not used in all VoxCeleb experiments. Vox1-O, Vox1-E, and Vox1-H are used as evaluation sets. The Vox1-O sets are used in the analysis part mainly, while the VoxCeleb1-E and Vox1-H are used to prove the generalisability and robustness of our model.

VoxCeleb1&2 and CnCeleb1&2 are used as the training sets of the CN series, which contains 7205 speakers and 2793 speakers, respectively. Two types of augmentation methods are adopted: (1) adding reverberation using RIR Noise dataset [36], (2) adding noise using MUSAN [37] dataset. CN-eval database is used as an evaluation set. The evaluation sets are described in Table 1.

**Table 1.** The detail of test datasets.

| Datasets | Spks | Utts | Trials |
| --- | --- | --- | --- |
| Vox1-O | 40 | 4708 | 37,611 |
| Vox1-E | 1251 | 145,160 | 579,818 |
| Vox1-H | 1190 | 137,924 | 550,894 |
| CN-eval | 196 | 17,973 | 3,484,292 |

### 6.2. Data Preprocessing

We use 64-dimensions FBanks as the raw acoustic features, which are extracted from 25 ms frames with 10 ms overlap, spanning the frequency range 0–8000 Hz. No voice activity detection (VAD) is applied.

### 6.3. Training Setting

As shown in Figure 2, the ResNet-50 architecture are used in our experiments. The initial number of channels is set as 32. To maintain a similar number of parameters with ResNet50, $t$ is set as 1.5 when $s$ is set as 8 in the following experiments according to Equation (3) in the HS-ResNet. Moreover, experiments show that inserting the DSSA module between stages 3 and 4 achieves better improvements. Thus, for convenience, the DSSA module is applied between stages 3 and 4 if not specially specified.

In the training stage, a mini-batch size of 32 is used to train models in all experiments. Softmax with cross-entropy loss is used to train models of the Vox series, and circle loss is used to train models of the Cn series if not specially specified. Stochastic gradient descent (SGD) with momentum 0.9, weight decay $1 \times 10^{-3}$ is used. The learning rate is set to 0.1, 0.01, and 0.001 and is switched when the training loss plateaus. Each speech sample in the training stage is sampled for L frames from each speech sample. The chunk size L is randomly sampled from the interval [L1; L2], and the interval is set to [200, 400], [300, 500], and [400, 600] in the three training stages.

### 6.4. Evaluation Metrics

In the testing stage, cosine similarity is applied as the back-end scoring method. The performance of different systems is gauged in terms of the EER and minDCF. We set the prior target probability to 0.01.

## 7. Results and Analysis

In this section, we compare and analyze three methods with baseline and other mainstream speaker verification methods in the same experimental setting. First, we present performance, EER, and minDCF, in three methods and analyze the complementarity between the three methods. In addition, we plot detection error trade-off (DET) on Vox1-H to further analyze performance. Then, class activation mapping (CAM) [38] is demonstrated for comparative analysis of three methods. Further, performance on CnCeleb is presented to further prove the generalisability and robustness of our proposed method. Finally, we compare the combination of three methods with other mainstream speaker verification methods.

### 7.1. Comparison and Analysis of Three Methods on the Vox Test Sets

We experiment with three methods on three test datasets, Vox1-O, Vox1-E, and Vox1-H. EER and minDCF are used as metrics. The training dataset of all experiments in this part only contains Vox2-dev without augmentation. As displayed in Table 2, three methods and their combinations are compared.

**Table 2.** Results of three methods and their combination on the Vox1-O/E/H.

| | Vox1-O | | Vox1-E | | Vox1-H | |
| | EER | minDCF | EER | minDCF | EER | minDCF |
| --- | --- | --- | --- | --- | --- | --- |
| ResNet50 | 2.03 | 0.196 | 1.86 | 0.209 | 3.28 | 0.319 |
| HS-Net50 | 1.46 | 0.136 | 1.34 | 0.163 | 2.58 | 0.243 |
| ResNet50+cross | 1.57 | 0.150 | 1.43 | 0.176 | 2.61 | 0.254 |
| ResNet50+DSSA | 1.76 | 0.157 | 1.55 | 0.173 | 2.84 | 0.264 |
| HS-Net50+cross | 1.26 | 0.104 | 1.22 | 0.144 | 2.39 | 0.226 |
| HS-Net50+DSSA | 1.27 | 0.101 | 1.19 | 0.147 | 2.29 | 0.219 |
| HS-Net50+cross+DSSA | 1.10 | 0.099 | 1.02 | 0.113 | 2.10 | 0.202 |

Firstly, we analyze the results of three methods. Three methods all achieve performance improvements. HS-ResNet50 exceeds the ResNet50 by 30% on EER and minDCF. The stronger ability to model long-range dependencies with HS-ResNet has proven that it is able to achieve great performance improvements through experiments. Its unique *split-stack* structure effectively collects more long-scale features and more long-range information. The cross-conv module achieves 20% improvements on EER and minDCF, compared with the no cros-conv method. By replacing the original 3 × 3 convolution kernel with a 1 × 5 and 5 × 1 convolution kernel, the RFs of the model are enlarged, and the ability of long-range feature extraction is enhanced. DSSA module achieves 15% improvements on EER and minDCF, which uses an explicit sparse attention strategy to capture effective long-range dependencies globally in each channel, and its parameters are increased just a little.

Then, we analyze the complementarity between the three methods. The HS-block and cross-conv focus on enlarging the local receptive field by stacking convolution filters and enlarging RFs in a single layer, respectively. The DSSA module focuses on integrating long-range information and extracting long-range context features globally. In theory, they are complementary to each other. Thus, we combined these methods into ResNet50. Compared with HS-Net50, HS-Net50+cross and HS-Net50+DSSA both achieve 15% improvements on EER and 25% improvements on minDCF, and HS-Net50+cross+DSSA achieve 25% improvements on EER and 35% improvements on minDCF. These demonstrate that there are strong complementarities between the three methods. The detection error trade-off on Vox1-H are shown in Figure 5. As can be seen from the figure, our proposed method achieves significant performance improvements.

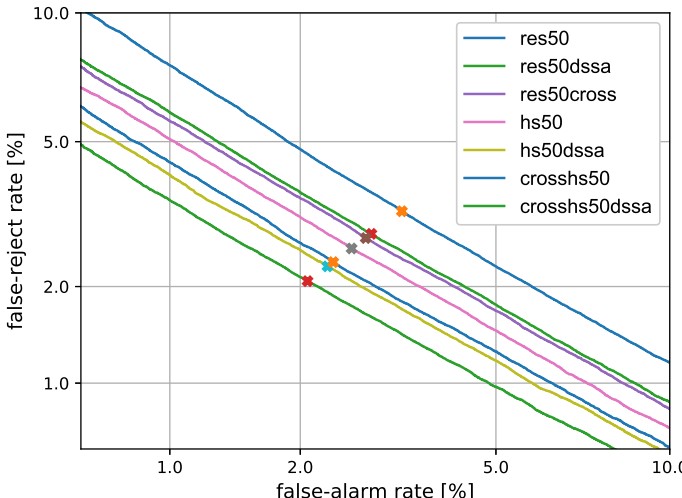

**Figure 5.** DET curves for all systems.

## 7.2. Comparison of Long-Range Feature Extraction Visually for Three Methods

Class activation mapping is used to demonstrate the impact of the three methods and combinations on speaker verification visually. We randomly select one utterance

in the VoxCeleb sets and perform the visualization analysis by Grad-CAM [39], which derives the attention weight distribution of the model on the feature map from the gradient information. The CAM heatmaps of three methods are shown in Figure 6. With these three approaches incorporated, the model tends to model long-time dependencies and extracts long-range features, which further demonstrates the effectiveness of our methods in extracting long-time features.

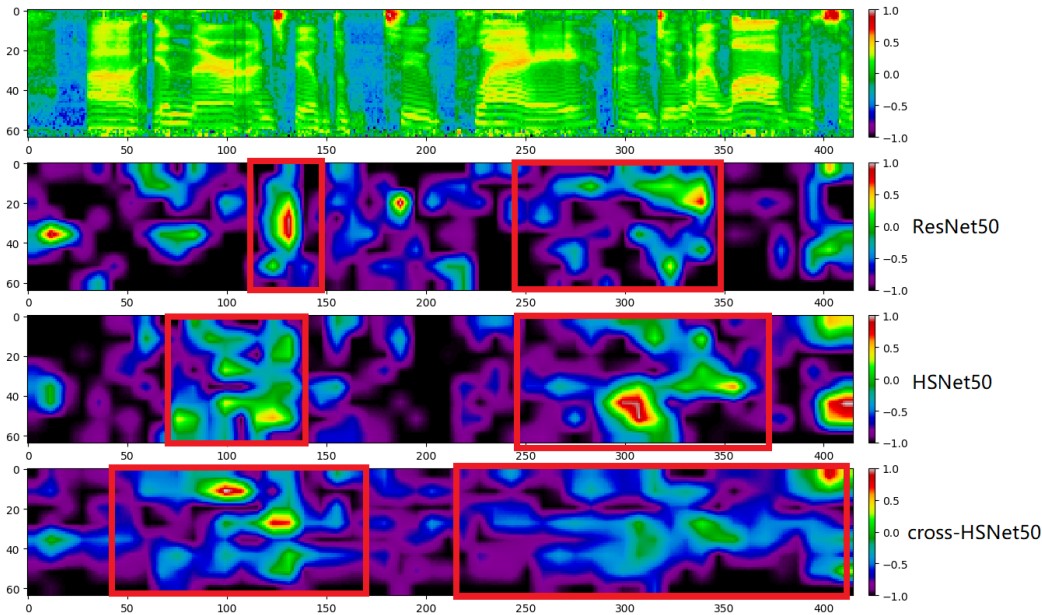

**Figure 6.** CAM heatmaps for three methods.

*7.3. Comparsion of Proposed Methods on the CnCeleb Dataset*

To further measure the generality and robustness of our method, we conduct experiments on CnCeleb sets. The experimental results are shown in Table 3 and are similar to the Voxceleb test. Each method and combination achieve performance improvements.

**Table 3.** Results of three methods and their combination on CN-eval.

| Training-Data | System | CN-Eval | |
|---|---|---|---|
| | | EER | minDCF |
| CN & Vox | ResNet50 | 6.89 | 0.398 |
| | ResNet50+DSSA | 6.20 | 0.376 |
| | ResNet50+cross | 6.46 | 0.388 |
| | HS-Net50 | 5.97 | 0.360 |
| | HS-Net50+cross | 5.62 | 0.369 |
| | HS-Net50+DSSA | 5.43 | 0.360 |
| | HS-Net50+cross+DSSA | 5.33 | 0.347 |
| | ResNet293+LM [40] | 5.23 | 0.316 |
| | RepVGG-A2 [41] | 5.85 | 0.293 |

*7.4. Comparison of Mainstream Methods on the Voxceleb Dataset*

To compare with other state-of-the-art models, we use another experimental setup for training our proposed methods on VoxCeleb2. Different from Section 6, we first use the SoX speed function with speeds 0.9 and 1.1 to generate extra twice speakers. Then, we use MUSAN and RIRs noises to perform online data augmentation. 80-dimensional FBank are used as input features instead of the original 64-dimensional Fbank. The training protocol is the same as [42]. The results are shown in Table 4. Our proposed methods achieve great performance.

**Table 4.** Comparison of EERs of our proposed model with others.

|  | Vox1-O | | Vox1-E | | Vox1-H | |
|---|---|---|---|---|---|---|
|  | EER | minDCF | EER | minDCF | EER | minDCF |
| ECAPA-TDNN [13] | 0.87 | 0.107 | 1.12 | 0.132 | 2.12 | 0.21 |
| SimAM-ResNet34 [43] | 0.64 | 0.067 | 0.84 | 0.089 | 1.49 | 0.146 |
| MBFA-MW-ECAPA-TDNN [44] | 0.87 | 0.115 | 1.22 | 0.135 | 2.31 | 0.222 |
| HS-Net50+cross+DSSA(ours) | 0.52 | 0.039 | 0.76 | 0.081 | 1.38 | 0.142 |

## 8. Conclusions

In this paper, we propose three methods to capture long-range dependencies and improve performance for speaker verification. The first method replaces the $3 \times 3$ convolution with an HS-block to enlarge the local RFs of a single layer. The second method proposes the DSSA module, which integrates information from the global perspective in each channel of the feature maps. The third method is the cross-convolution kernel, which replaces the traditional $3 \times 3$ convolution kernel with a $1 \times 5$ and $5 \times 1$ convolution kernel. These methods capture long-range dependencies from different perspectives and are highly complementary. To evaluate the robustness of our methods, we conduct experiments in VoxCeleb and CnCeleb sets. By integrating them into one model, we achieve state-of-the-art performance in VoxCeleb sets and CnCeleb sets.

**Author Contributions:** Conceptualization, Z.L.; methodology, Z.L.; software, Z.L. and Z.Z.; validation, Z.L.; formal analysis, Z.L.; investigation, Z.L.; resources, Z.L.; data curation, Z.L., W.W. and P.Z.; writing—original draft preparation, Z.L.; writing—review and editing, Z.L., W.W., P.Z. and Q.Z.; visualization, Z.L.; supervision, W.W. All authors have read and agreed to the published version of the manuscript.

**Funding:** This research received no external funding.

**Institutional Review Board Statement:** Not applicable.

**Informed Consent Statement:** Not applicable.

**Data Availability Statement:** Publicly available datasets are analyzed in this study. This data can be found here: [https://www.robots.ox.ac.uk/vgg/data/voxceleb/index.html, accessed on 1 March 2018; http://cnceleb.org/, accessed on 10 April 2020].

**Conflicts of Interest:** The authors declare no conflict of interest.

## Abbreviations

The following abbreviations are used in this manuscript:

| | |
|---|---|
| HS-Block | Hierarchical-split block |
| cross-conv | cross convolution kernel |
| RFs | receptive fields |
| DSSA | Depthwise Separable Self-Attention |
| SV | speaker verification |
| TDNN | Time Delay Neural Network |
| CNN | Convolutional Neural Network |
| ECAPA-TDNN | Emphasized Channel Attention, Propagation, and Aggregation in TDNN |
| LSTM | Long Short Term Memory |
| CV | Computer Vision |
| spks | speakers |
| utts | utterances |

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
