# Peer review of "Explore Long-Range Context Features for Speaker Verification"

_applsci, doi:10.3390/app13031340_

Round 1
Reviewer 1 Report
The contributions of the work are as follows:
- The authors proposed a cross-Conv to enlarge RFs locally.
- The authors proposed a DSSA module to improve performance.
My concern is as follows:
- What back-end is used in Section Experiments?
- Please add Section Related Works.
- The authors should briefly introduce Stage 1, Stage 2, Stage 3, and Stage 4 in Figure 1.
- The author said that the proposed DSSA module is a plug-and-play module. Is it a pre-trained module? How to train the module?
- Equation (3) is not straightforward. Please give a detailed introduction.
- What is the $d_k$ in Equation (4)?
- Why did the authors select 64-dimensional FBanks? How to choose these 64 FBank coefficients?
- Why are the work’s results in Table 4 better than the results of the HS-Net50+cross+DSSA in Table 2?
- Some references are not written correctly, such as 24 and 37.
Reviewer 2 Report
There were proposed 3 methods for advancing speaker verification features extraction in the paper. They include the hierarchical-split block, the cross convolutional kernel, and the Depthwise Separable Delf-Attention module.
Despite that results are promising the paper needs corrections.
Main point.
177 line. “Following the previous work in [4], the ResNet-50 architecture are used in our experiments.” The referenced paper "Exploring the Encoding Layer and Loss Function in End-to-End Speaker and Language Recognition System" has no description of ResNet-50 architecture. Without it the baseline system is undefined. As an original Resnet-50 model is devoted for image processing, it remains unclear how the aggregation of time frames to a single embedding is performed. A good options is to give reference to source code of model.
Some minor points
In non-numbered lines between 110-111 “After the 1 ∗ 1 convolution, the feature maps are split equally into s groups”. In the Figure1(b) we see a split into groups before 1 * 1 convolution.
112-113 lines. “t means how many times the number of channels will be expanded”. Parameter t needs more detailed explanation. Where we can see the expansion in Figure 1(b)?
117 line. What is “h”?
89 line. Typo “Thrid, we propose …”
Round 2
Reviewer 1 Report
Although the authors have revised and improved the manuscript, the revision is still imperfect.
1. In Section Related works, some works related to speaker verification need to be further supplemented.
2. The authors must explain in detail how to extract the 64 FBank coefficients rather than give a python package. Generally, the extracted FBank is the 80-dimensional feature, and the authors also use the 80-dimensional FBank features in Section 7.4. Therefore, the authors must explain how to select 64-dimensional features from the original 80 FBank coefficients.
3. Since the results in Table 4 are better than those in Table 2, the methods used in Section 7.4 are better than the proposed method. So why not use a better method (the method used in Section 7.4) directly?
4. The author's explanation of the back end is not sufficient. The authors need to describe the outputs of the proposed model in detail and explain how to use the model's output to rate the speaker to be verified with the Cosine similarity.
5. The authors need to improve the experimental part further and explain whether the model's input is 1D or 2D. How to feed the features into the model? What is the output of the model? According to the Cross evolution kernel in Section 5, the author should have adopted 2D input. How to organize the 64-dimensional FBank features into 2D input?
6. References 24, 26, and 37 are conference papers, and the authors have not thoroughly proofread the references in the revised version.
